# Layer- and cell type-selective co-transmission by a basal forebrain cholinergic projection to the olfactory bulb

Daniel T. Case[1,2], Shawn D. Burton [2,3,4], Jeremy Y. Gedeon[1], Sean-Paul G. Williams[1], Nathaniel N. Urban[1,2,3] & Rebecca P. Seal [1,2]

Cholinergic neurons in the basal forebrain project heavily to the main olfactory bulb, the first processing station in the olfactory pathway. The projections innervate multiple layers of the main olfactory bulb and strongly influence odor discrimination, detection, and learning. The precise underlying circuitry of this cholinergic input to the main olfactory bulb remains unclear, however. Here, we identify a specific basal forebrain cholinergic projection that innervates select neurons concentrated in the internal plexiform layer of the main olfactory bulb. Optogenetic activation of this projection elicits monosynaptic nicotinic and GABAergic currents in glomerular layer-projecting interneurons. Additionally, we show that the projection co-expresses markers for GABAergic neurotransmission. The data thus implicate neurotransmitter co-transmission in the basal forebrain regulation of this inhibitory olfactory microcircuit.

[1] Department of Neurobiology, University of Pittsburgh School of Medicine, Pittsburgh, PA 15213, USA. [2] Center for the Neural Basis of Cognition, Pittsburgh, PA 15213, USA. [3] Department of Biological Sciences, Carnegie Mellon University, Pittsburgh, PA 15213, USA. [4]Present address: Department of Neurobiology and Anatomy, University of Utah, Salt Lake City, UT 84112, USA. Correspondence and requests for materials should be addressed to R.P.S. (email: rpseal@pitt.edu)

Cholinergic neurons within basal forebrain (BF) nuclei project to many different brain regions where they have an important role in learning and memory, attention, and cognition. One BF nucleus, the horizontal diagonal band of Broca (HDB), densely innervates the main olfactory bulb (MOB), the first processing station within the main olfactory system[1–4]. Cholinergic projections to the MOB are most heavily concentrated in the glomerular layer (GL) and internal plexiform layer (IPL), with sparser projections to other layers[1–4], and have been implicated in odor discrimination and detection as well as olfactory learning[5–8]. The precise functional connectivity of these projections within the MOB remains controversial, however. Pharmacological investigation of cholinergic signaling among mitral and tufted cells (M/TCs), the principal neurons of the MOB, has provided evidence both for[4] and against[9, 10] direct muscarinic inhibition of M/TCs, in addition to direct nicotinic excitation[4, 9–12]. Likewise, electrical or optogenetic stimulation of cholinergic neurons in the HDB in vivo has triggered depression[13], facilitation[14–16], and mixed facilitation-depression[7] of spontaneous M/TC firing. Further, enhancing cholinergic signaling in the MOB can sharpen M/TC odor tuning[8, 17] or add an excitatory bias to M/TC odor responses without affecting tuning[7]. Despite these strong though conflicted effects of cholinergic signaling on M/TC activity, MCs strikingly exhibit no postsynaptic response to brief optogenetic stimulation of cholinergic projections in vitro[10] and show only an ~ 0.5 mV hyperpolarization with a 15 s-long stimulation[4]. Similarly conflicting results have emerged for the cholinergic modulation of granule cells (GCs), the most abundant GABAergic interneurons in the MOB[4, 8, 9, 18, 19]. Underscoring this controversy and the lack of clarity on the functional role of cholinergic projections is the potential for more complex neurotransmitter signaling. Indeed, we and others have previously reported that a subset of BF cholinergic neurons express the vesicular glutamate transporter 3 (VGLUT3), one of three transporters responsible for packaging glutamate into synaptic vesicles for regulated release[20–22].

Here, we show that a similar population of VGLUT3+ cholinergic neurons is also present in the HDB of adult mice. Targeting channelrhodopsin-2 (ChR2) expression to this subset of BF neurons, we show that their projections are highly enriched in the IPL, where they selectively innervate deep short-axon cells (dSACs). This finding suggests that subpopulations of BF neurons differentially modulate MOB circuits. Surprisingly, while we do not detect significant glutamatergic postsynaptic currents when stimulating these projections in adult mice, we do detect robust monosynaptic GABAergic currents that are elicited simultaneously with monosynaptic nicotinic currents. Further support for GABA co-transmission is provided by showing the colocalization of cholinergic and GABAergic markers within HDB neurons as well as their axonal projections to the IPL. These data thus demonstrate that the regulation of target neurons in the MOB by BF cholinergic projections is neurochemically more complex than originally anticipated.

## Results

### VGLUT3+ cholinergic neurons in the HDB project to the IPL.
To determine whether VGLUT3 is expressed by cholinergic neurons in the HDB, we crossed VGLUT3Cre mice[23] to the Cre-dependent tdTomato Ai14 reporter line[24] and immunostained for the vesicular acetylcholine transporter (VAChT). We found that a significant percentage of VAChT+ neurons expressed tdTomato (46%), indicating that cholinergic neurons in this nucleus also express or have expressed VGLUT3 at some point during development (Fig. 1a–c). Virtually all tdTomato+ neurons

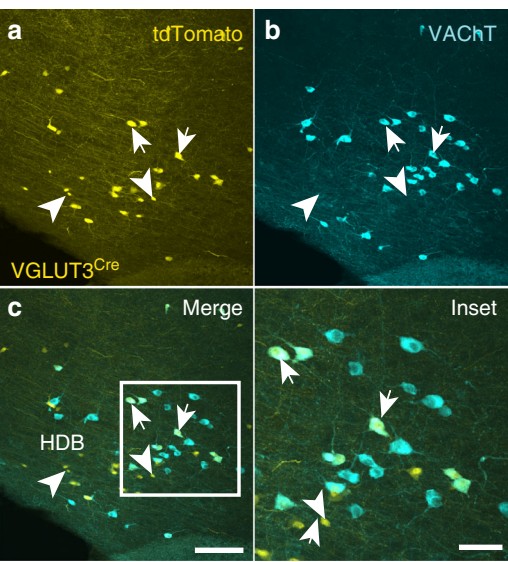

**Fig. 1** HDB cholinergic neurons express tdTomato in VGLUT3Cre-tdTomato mice. **a–c** HDB from a P56 VGLUT3Cre-tdTomato mouse shows considerable overlap between VAChT (*cyan*) and the tdTomato reporter (*yellow*). Approximately 46% of VAChT+ cells are positive for tdTomato and ~48% of tdTomato+ cells express VAChT (108 of 237 cells and 59 of 121 cells, respectively, n = 3 animals). *Arrows* denote colocalization; *arrowheads* denote tdTomato without VAChT. *Scale bar* = **a–c** 100 μm, (**c** inset) 20 μm

with large somata (21 ± 2 μm) were VAChT+ (97%, 58/60 cells) while a subpopulation of tdTomato+ neurons having small somata (11 ± 1 μm) was VAChT− (Fig. 1a–c). To investigate the connectivity of HDB cholinergic projections to MOB neurons, and whether they release glutamate in addition to acetylcholine, we injected a virus encoding ChR2 fused to the enhanced yellow fluorescent protein (AAV9-EF1a-DIO-ChR2-EYFP) in the HDB of VGLUT3Cre mice (Fig. 2a). Strikingly, the expression of ChR2-EYFP in axonal projections to the MOB was highly enriched in the IPL, with sparser innervation observed in other layers (Fig. 2b, c, f, Supplementary Figs. 1 and 2). In contrast, injection of the virus into the HDB of choline acetyltransferase (ChAT)Cre mice[25] (Fig. 2d) yielded the expected distribution for ChR2-EYFP in axonal projections predominantly to the GL and IPL, as well as sparser labeling in other layers (Fig. 2e,f; Supplementary Fig. 1)[1–4]. Consistent with expression in cholinergic neurons, ChR2-EYFP in the MOB of VGLUT3Cre mice colocalized with VAChT immunoreactivity (Fig. 3a). Similarly, colocalization of tdTomato and VAChT was observed in the IPL of VGLUT3Cre-tdTomato mice (Fig. 3b). Enrichment of ChR2-EYFP+ projections in the IPL of VGLUT3Cre mice was not due to discrepancies in viral infection between VGLUT3Cre and ChATCre mice, as 56% of the target population was consistently infected in each mouse line (Fig. 2a, d). Our results therefore suggest that cholinergic neurons projecting to the MOB are composed of at least two spatially intermingled neuronal populations within the HDB: VGLUT3+ neurons that preferentially innervate the IPL, and VGLUT3− neurons that innervate the GL and likely also the IPL.

### Cell type-selective innervation by VGLUT3+ cholinergic neurons.
The observation of specific subsets of HDB neurons forming distinct projections to the MOB raised questions about the circuit properties of these different neuronal projections. We therefore performed whole-cell voltage-clamp recordings in the MOB while optogenetically activating HDB projections in acute

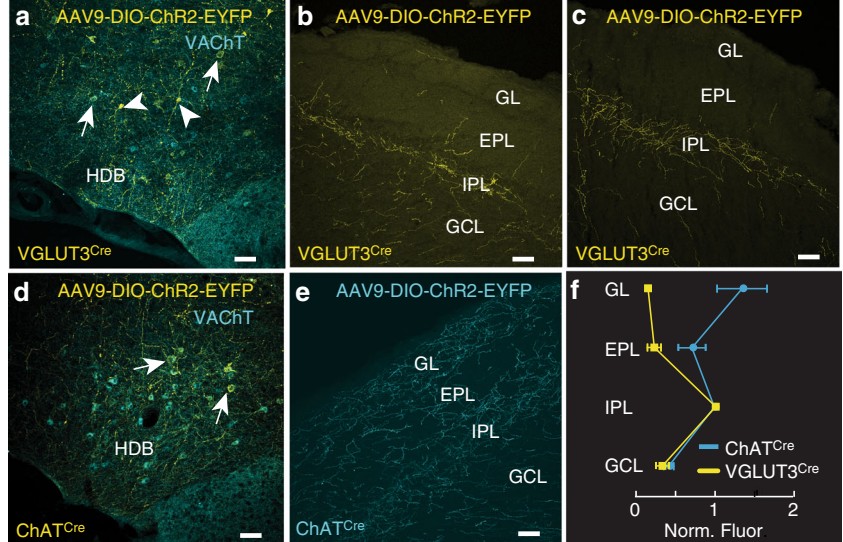

**Fig. 2** VGLUT3[+] neurons in the HDB preferentially innervate the IPL of the MOB. **a** Approximately 26% (61 of 238 cells, *n* = 6 animals) of the HDB neurons that express VAChT (*cyan*) were infected following virus injection (*yellow*) in VGLUT3[Cre] mice. **b**, **c** Injection of ChR2-EYFP virus into the HDB of VGLUT3[Cre] mice labels axons preferentially innervating the IPL. Representative images from two mice are shown. **d** Approximately 56% (47 of 84 cells, *n* = 2 animals) of the HDB neurons that express VAChT (*cyan*) were infected following ChR2-EYFP virus injection (*yellow*) into ChAT[Cre] mice. **e** ChR2-EYFP virus injected into the HDB of ChAT[Cre] mice labels axons densely innervating both the GL and IPL. Given that ~46% of cholinergic neurons are VGLUT3[+] (Fig. 1), the numbers indicate that ~56% of neurons positive for both VGLUT3 and VAChT were infected with the virus, consistent with what is observed in the ChAT[Cre] animals. **f** Quantification of ChR2-EYFP-labeled axons in the two mouse lines shows a peak in the IPL in VGLUT3[Cre] mice (*yellow*) and peaks in the GL and IPL in ChAT[Cre] mice (*cyan*). *Scale bars* = 100 μm

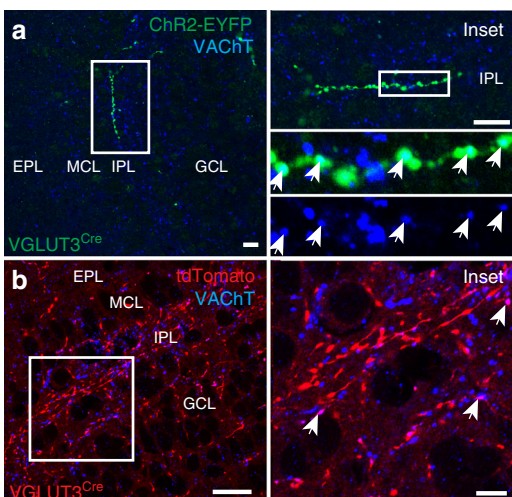

**Fig. 3** VGLUT3[+] HDB projections to the IPL express VAChT. **a** ChR2-EYFP-labeled projections to the IPL (*green*) in a HDB-injected VGLUT3[Cre] mouse colocalize with VAChT (*blue*). **b** VGLUT3[Cre]-tdTomato mouse shows colocalization of VAChT (*blue*) and tdTomato (*red*) in the IPL of the MOB. *Scale bars* = **a**, **b** 10 μm; **b** inset 5 μm

slices prepared from virus injected VGLUT3[Cre] and ChAT[Cre] mice. Recordings were performed ~2 weeks after injection (postnatal days 35–47; Fig. 4a, b). Consistent with previous results[4, 10], brief light pulses (10–20 ms) did not evoke significant inward currents ($V_{hold}$ = −60 mV) in MCs in either mouse line (Fig. 5). We likewise observed no significant input to numerous other MOB cell types, including TCs, GCs, and external plexiform layer interneurons (EPL-INs; Fig. 5). Importantly, however, 19 of 22 dSACs having cell bodies in or near the IPL exhibited robust light-evoked short-latency excitatory postsynaptic currents

(EPSCs) (Figs. 4c–f and 5). The peak EPSC amplitudes did not significantly differ between ChAT[Cre] and VGLUT3[Cre] mouse lines (Fig. 4h), consistent with the idea that the VGLUT3[+] HDB neurons form a large proportion of the cholinergic projections from the HDB to the IPL. Repetitive optogenetic activation at 5 and 20 Hz evoked EPSCs with high probability (Supplementary Fig. 3), suggesting that bursts of HDB activity may powerfully influence dSAC activity.

MOB dSACs can be subdivided by their axonal projections into granule cell layer (GCL)-, EPL-, and GL-projecting dSACs (GCL/EPL/GL-dSACs)[26]. Strikingly, post hoc morphological reconstruction of the recorded dSACs revealed that BF input preferentially (and possibly exclusively) targeted GL-dSACs (Fig. 5). Of the 22 total dSACs recorded, we recovered 9 with sufficiently intact axons to enable classification: 6 were GL-dSACs, 0 were EPL-dSACs, and 3 were GCL-dSACs, consistent with the laminar distribution of GL-, EPL-, and GCL-dSACs previously observed in rats[26]. Light-evoked EPSCs were detected in all 6 confirmed GL-dSACs, but not in any of the confirmed GCL-dSACs (Fig. 5o). Moreover, the 13 dSACs with unrecovered axonal projection patterns (?-dSACs) all exhibited light-evoked EPSCs and were located in the IPL or superficial GCL, where GL-dSACs comprise the majority of dSACs[26, 27]. Given the limited number of recovered GCL- and EPL-dSACs, and the focus of our recordings to cells in or near the IPL where dense HDB projections were observed, we cannot exclude the possibility that HDB projections likewise innervate EPL-dSACs and/or a subset of GCL-dSACs. Nevertheless, our data strikingly demonstrate that cholinergic HDB projections to the IPL selectively innervate dSACs, and preferentially (and possibly exclusively) innervate GL-dSACs.

**Co-transmission by an HDB projection to the IPL.** The fact that VGLUT3[+] HDB projections to the IPL express VAChT (Fig. 3) suggests that the light-evoked EPSCs recorded in dSACs may be cholinergic, glutamatergic, or both. We therefore performed

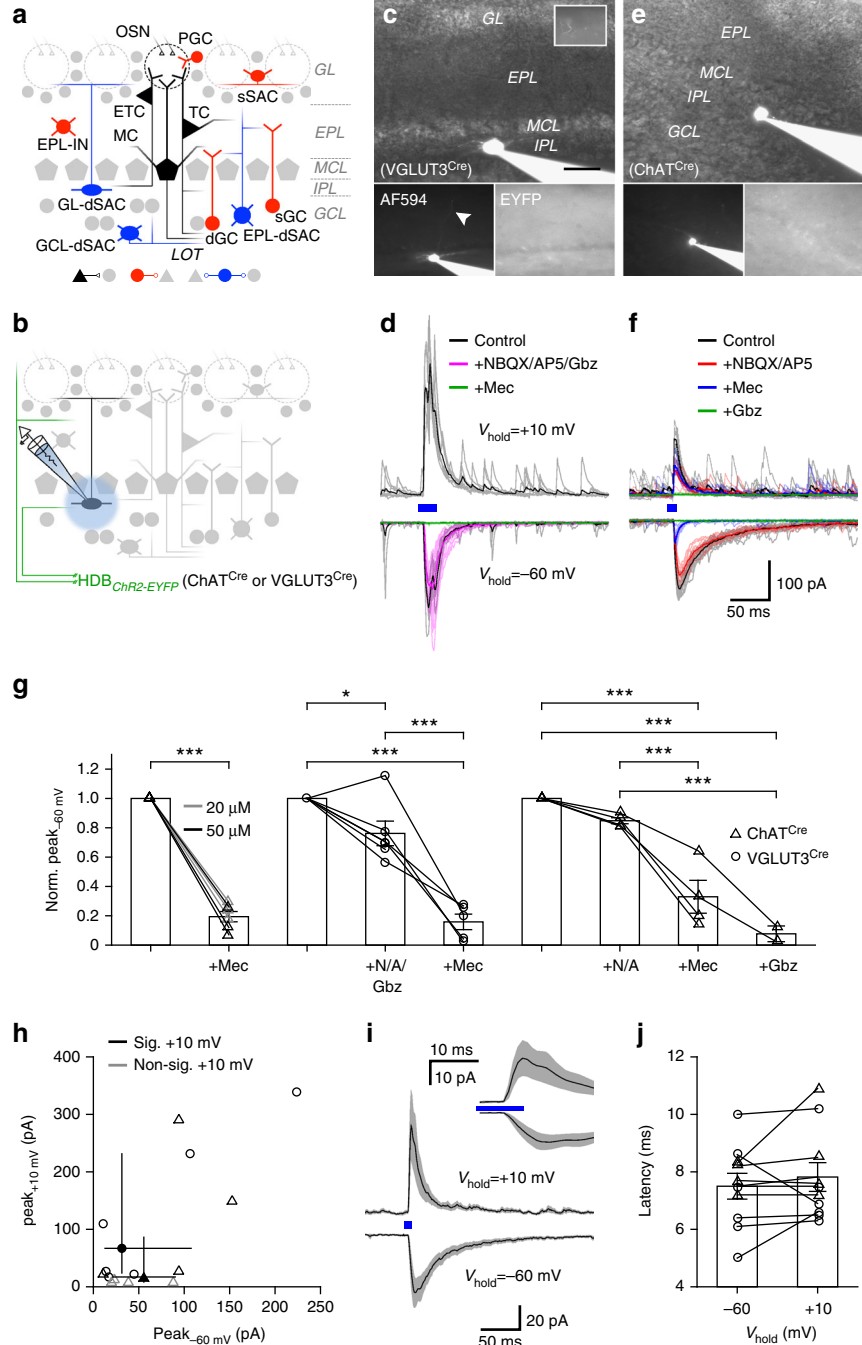

**Fig 4** HDB projections to the IPL mediate mixed nicotinic-GABAergic postsynaptic currents in dSACs. **a** MOB circuitry. **b** Experimental design. **c**, **d** Representative GL-dSAC recording (*arrowhead*: axon; *inset*: axon in GL; *scale bar*: 20 µm) from a VGLUT3$^{Cre}$ mouse. Optogenetic stimulation evoked EPSCs and IPSCs at negative and positive holding potentials, respectively. NBQX/AP5/Gbz modestly reduced light-evoked EPSCs. Subsequent mecamylamine (Mec) abolished the residual EPSC. Mean response shown; individual trials in lighter color. **e**, **f** Same as **c**, **d** for a ?-dSAC of a ChAT$^{Cre}$ mouse. NBQX/AP5 modestly reduced the light-evoked EPSC. Subsequent Mec strongly reduced the residual EPSC, but not IPSC. Subsequent Gbz abolished the residual IPSC (and unclamped component at $V_{hold} = -60$ mV). **g** Summary of EPSC pharmacology. In ChAT$^{Cre}$ mice (*left*), Mec reduced EPSC amplitudes to $19.4 \pm 8.6\%$ of control values ($n = 6$; $p = 2.9 \times 10^{-6}$, two-tailed paired $t$-test). In VGLUT3$^{Cre}$ mice (*middle*), NBQX/AP5/Gbz reduced EPSC amplitudes to $76.2 \pm 20.5\%$ ($n = 6$); subsequent Mec reduced EPSC amplitudes to $15.8 \pm 11.8\%$ ($n = 5$) ($p = 7.4 \times 10^{-8}$, one-way ANOVA; control vs. NBQX/AP5/Gbz, $p = 0.015$, control vs. Mec, $p = 5.4 \times 10^{-8}$, NBQX/AP5/Gbz vs. Mec, $p = 5.6 \times 10^{-6}$, post hoc Tukey-Kramer). In ChAT$^{Cre}$ mice (*right*), NBQX/AP5 insignificantly reduced EPSC amplitudes ($p = 0.38$), while subsequent Mec and Gbz reduced EPSC amplitudes to $33.0 \pm 22.4\%$ ($n = 4$) and ~ $7.7\%$ ($n = 2$), respectively ($p = 1.2 \times 10^{-5}$, one-way ANOVA; control vs. Mec, $p = 1.1 \times 10^{-4}$, control vs. Gbz, $p = 3.9 \times 10^{-5}$, NBQX/AP5 vs. Mec, $p = 8.7 \times 10^{-4}$, NBQX/AP5 vs. Gbz, $p = 1.8 \times 10^{-4}$, post hoc Tukey-Kramer). **h** Peak EPSC and IPSC amplitudes. *Closed symbols* (and *error bars*): median and first/third quantiles). EPSC and IPSC amplitudes did not differ between the two mouse lines (EPSC: $57.0 \pm 42.7$ [$n = 13$] vs. $69.7 \pm 83.4$ pA [$n = 6$], ChAT$^{Cre}$ vs. VGLUT3$^{Cre}$, $p = 0.90$, two-sided Wilcoxon rank-sum test; IPSC: $65.1 \pm 103.3$ [$n = 8$] vs. $124.5 \pm 134.3$ pA [$n = 6$], ChAT$^{Cre}$ vs. VGLUT3$^{Cre}$, $p = 0.18$, two-sided Wilcoxon rank-sum test). ChAT$^{Cre}$ and VGLUT3$^{Cre}$ data are therefore pooled for subsequent analyses. **i** Mean EPSC ($n = 18$) and IPSC ($n = 14$) waveforms. *Inset*: expanded timescale. **j** EPSC and IPSC latencies did not differ ($7.5 \pm 1.4$ vs. $7.8 \pm 1.6$ ms; EPSC vs. IPSC; $n = 10$; $p = 0.39$, two-tailed paired $t$-test)

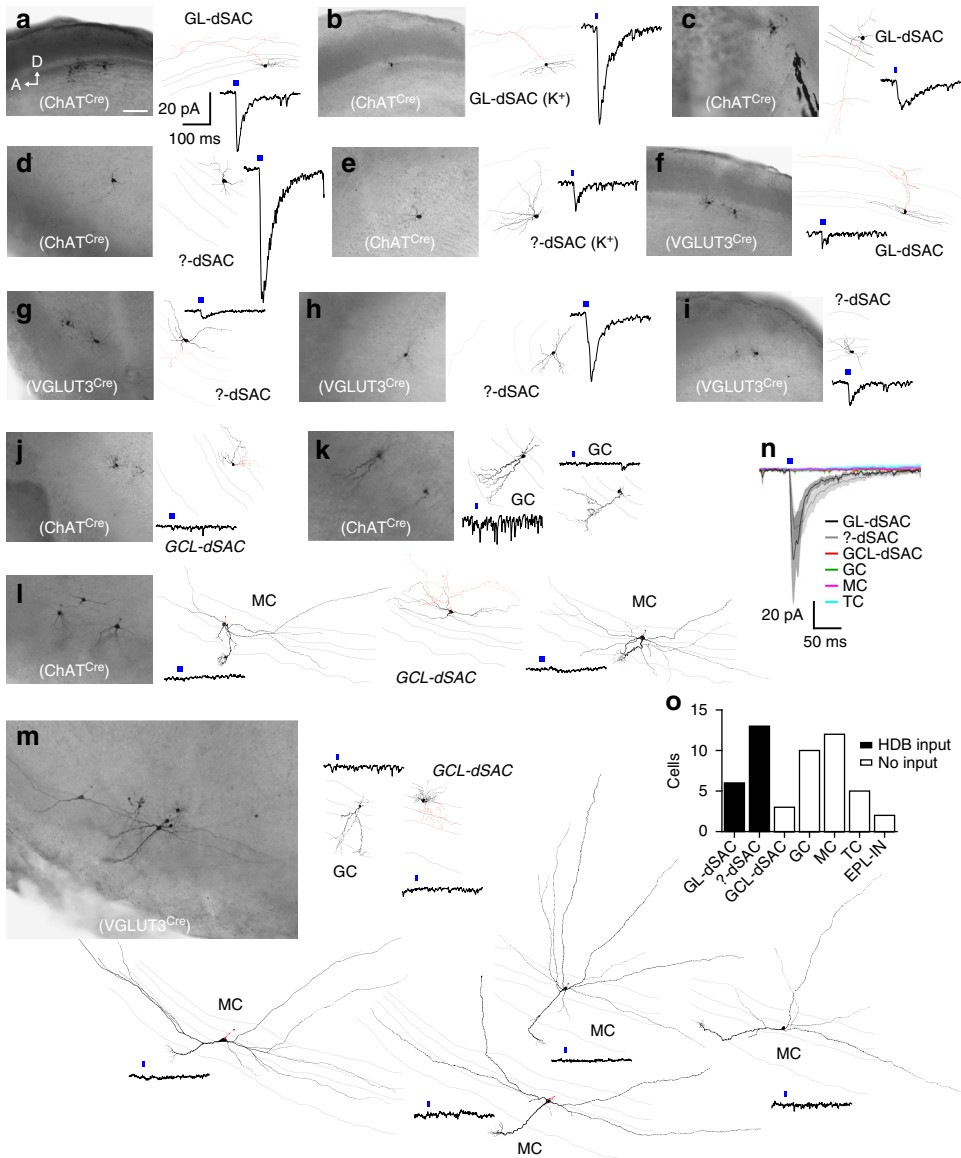

**Fig. 5** HDB projections to the IPL innervate GL-dSACs. **a-m** Post hoc Neurobiotin brightfield image (*scale bar*: 100 μm), morphological reconstruction, and mean current ($V_{hold} = -60$ mV) of a subset of recorded MOB cells exhibiting significant **a-i** or non-significant **j-m** responses to brief optogenetic activation (*blue line*) of BF projections in ChAT^Cre or VGLUT3^Cre mice. Gray lines in reconstructions denote GL/EPL, EPL/MCL, MCL/IPL, and IPL/GCL borders. A subset of cells **b**, **e** were recorded with a K^+-based intracellular solution. A, anterior; D, dorsal. **n** Mean light-evoked responses ($V_{hold} = -60$ mV) across GL-dSACs ($n = 6$), ?-dSACs ($n = 13$), GCL-dSACs ($n = 3$), GCs ($n = 9$), MCs ($n = 11$), and TCs ($n = 3$). **o** Total number of different MOB cell types exhibiting a significant response ($V_{hold} = -60$ mV) to optogenetic activation of HDB projections

a series of pharmacological experiments to determine the identity of the neurotransmitters released at these synapses. First, we tested whether the VGLUT3^+ projections release acetylcholine. Light-evoked EPSCs persisted following combined application of glutamatergic antagonists NBQX (10 μM) and AP5 (50 μM) and GABAergic antagonist gabazine (Gbz; 10 μM), but were abolished by subsequent application of the nicotinic antagonist mecamylamine (Mec; 20–50 μM; Fig. 4c, d, g). Mec likewise strongly reduced the light-evoked EPSC in slices from ChAT^Cre mice (Fig. 4g). Subtraction of post-Mec currents from pre-Mec currents across both mouse lines yielded EPSCs with kinetics ($\tau = 33.5 \pm 19.6$ ms, $n = 16$) consistent with α2 subunit-containing nicotinic receptors[28], providing further evidence for HDB innervation of GL-dSACs[27]. We did not observe any evidence of long-lasting Mec-insensitive currents mediated by muscarinic receptors following brief light pulses (10–20 ms) in either dSACs or other MOB cell types (Figs. 4d, f and 5n), including in a subset of cells recorded with a K^+-based intracellular solution (Fig. 5b, e). The lack of light-evoked muscarinic currents is consistent with a recent report in which no muscarinic modulation of MCs was observed following brief optogenetic stimulation of HDB cholinergic projections[10], and only very weak hyperpolarization (~0.5 mV) was observed following prolonged (10 Hz train over 15 s) optogenetic stimulation[4]. Collectively, our results demonstrate that VGLUT3^+ HDB neurons project preferentially to the IPL and release acetylcholine to mediate robust nicotinic input onto dSACs. Of note, this is the first reported observation of monosynaptic cholinergic currents in the MOB.

We next examined whether HDB projections to the IPL release glutamate. To avoid the possible confound of optogenetically activating the non-cholinergic subpopulation of

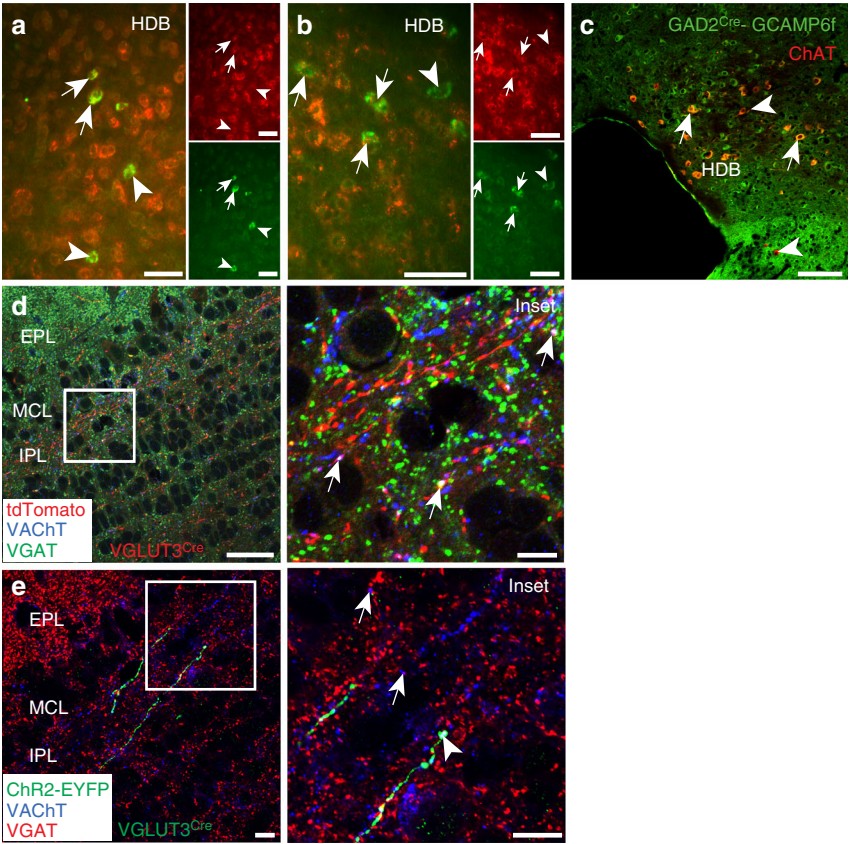

**Fig. 6** Colocalization of markers for acetylcholine and GABA in the HDB and MOB. **a** HDB from VGLUT3^Cre-tdTomato mouse shows *tdTomato* mRNA (*green*) partially colocalizes with *VGAT* mRNA (*red, arrows*). Not all *tdTomato* colocalizes with *VGAT* (*arrowheads*). **b** HDB from VGLUT3^Cre-tdTomato mouse shows *VAChT* mRNA (*green*) partially colocalizes with *VGAT* mRNA (*red, arrows*). Not all *VAChT* colocalizes with *VGAT* mRNA (*arrowheads*). **c** HDB from GAD2^Cre-GCAMP6f mouse shows colocalization of ChAT (*red*) and GCAMP6f (*green, arrows*). *Arrowhead* marks a ChAT^+ neuron not colocalized with GCaMP6f. **d** MOB from a VGLUT3^Cre-tdTomato mouse shows colocalization of VGAT (*green*) and VAChT (*blue, arrows*). *Inset* from **d** is the same as in Fig. 3b but also with VGAT (*green*). tdTomato, VGAT, and VAChT colocalize (*arrows*). **e** MOB from VGLUT3^Cre mouse injected with ChR2-EYFP virus in the HDB shows colocalization of VGAT (*red*) and VAChT (*blue, arrows*). VGAT and VAChT colocalize with a ChR2-EYFP-labeled axon (*green, arrowhead*). *Scale bars* = **a–c** 50 μm, **d** 20 μm, **d** inset 5 μm, **e** 10 μm, **e** inset 10 μm

VGLUT3^+ HDB neurons (Fig. 1), we used ChAT^Cre mice. Application of NBQX/AP5 consistently reduced the peak EPSC amplitudes to $84.8 \pm 4.1\%$ of baseline (range: 81.1–90.0%), though this effect was not significant ($p = 0.38$; $n = 4$; Fig. 4f, g). Interestingly, a residual light-evoked post-synaptic current was observed even in the presence of glutamatergic and nicotinic antagonists (Fig. 4f, g). We therefore tested whether this residual postsynaptic current reflected an incompletely voltage-clamped GABAergic input. In 10 of the 14 cells tested that exhibited light-evoked EPSCs, depolarization to +10 mV (i.e., near the EPSC reversal potential) revealed a pronounced light-evoked inhibitory postsynaptic current (IPSC) (Fig. 4d, f, h). The latencies of the light-evoked EPSCs and IPSCs did not differ (Fig. 4i, j), suggesting that the two neurotransmitters are released simultaneously by ChR2-EYFP^+ HDB projections and arguing against the possibility that the IPSCs reflect feedforward inhibition. Consistent with the direct release of both acetylcholine and GABA rather than feedforward inhibition, light-evoked IPSCs persisted after excitatory input was blocked (with NBQX/AP5 and Mec), but were abolished by subsequent application of Gbz (Fig. 4f, g). In total, 6 of 6 cells in VGLUT3^Cre mice and 4 of 8 cells in ChAT^Cre mice exhibited light-evoked IPSCs (Fig. 4h). Results from our pharmacological experiments thus demonstrate that HDB cholinergic neurons projecting to the IPL release both acetylcholine and GABA onto dSACs.

**Colocalization of GABAergic and cholinergic markers**. To further confirm the GABAergic nature of cholinergic HDB projections to the MOB, we performed double fluorescent in situ hybridization for the *vesicular GABA transporter* (*VGAT*) and *VAChT* transcripts as well as for *VGAT* and *tdTomato* transcripts in HDB slices from adult VGLUT3^Cre-tdTomato mice (Fig. 6a, b). Consistent with our electrophysiological results demonstrating release of both acetylcholine and GABA, 47% of *VAChT*^+ cells (24 of 51 total, $n = 5$ slices from 2 animals) and 43% of *tdTomato*^+ cells (34 out of 79 total cells, $n = 8$ slices from 2 animals) expressed *VGAT* transcript. We also immunostained the HDB of glutamic acid decarboxylase 2 (GAD2)^Cre mice[29] crossed to Cre-dependent GCaMP6f reporter mice[30] for green fluorescent protein (GFP) and ChAT (Fig. 6c). Consistent with our double fluorescent in situ hybridization results, 48% of the ChAT^+ neurons (147 out of 305 cells, $n = 5$ slices from one animal) expressed GCaMP6f. Lastly, we investigated whether markers for these neurotransmitters colocalize in the IPL. Immunostaining the MOB of adult VGLUT3^Cre-tdTomato mice for VAChT and VGAT revealed punctate colocalization of VAChT, VGAT, and tdTomato (Fig. 6d), albeit to a lesser extent than colocalization of VAChT with tdTomato alone (Fig. 3b). ChR2-EYFP^+ projections in the IPL of HDB-injected VGLUT3^Cre mice showed colocalization of VAChT with ChR2-EYFP, and VAChT with VGAT, as well as sparser observations of all three markers together (Fig. 6e). Taken together, our data indicate that

VGLUT3[+] HDB neurons form a subset of cholinergic projections to the MOB IPL that synapse onto dSACs and release both acetylcholine and GABA.

## Discussion

Here, we have shown that multiple distinct populations of BF cholinergic neurons in the HDB project to the MOB. One population is labeled in VGLUT3[Cre] mice and preferentially innervates the IPL, while the other population densely innervates the GL and more sparsely the other layers, likely including the IPL. We have further shown that VGLUT3[+] projections selectively innervate dSACs (and possibly exclusively GL-dSACs) and signal with a more diverse array of neurotransmitters than expected (GABA in addition to acetylcholine), thus allowing for a wider range of signaling mechanisms and functional outcomes.

Our results have several broad implications. Foremost, we provide the first direct electrophysiological evidence for monosynaptic cholinergic input from the HDB to the MOB. Moreover, the discovery of GL-dSACs as a target of HDB cholinergic projections to the IPL points to an unexpected and highly specific mechanism of neuromodulation in the MOB. Each GL-dSAC axon arborizes across several tens of glomeruli[26, 27] and potently inhibits principal tufted cell output[27]. HDB regulation of GL-dSAC activity is thus likely critical in coordinating sensory processing across widespread regions of the MOB, consistent with cholinergic modulation of odor discrimination and detection and olfactory learning[5–8]. These findings also suggest that other subsets of HDB neurons (e.g., VGLUT3[−] neurons) may in turn be targeting other elements of the MOB circuit. Previous studies of HDB projections have used approaches that simultaneously activated projections to multiple layers of the MOB[4, 5, 7, 8, 13–16, 18], and thus likely resulted in mixed effects on MOB activity. Our data suggest that understanding the overall role of HDB projections to the MOB will ultimately require employing new tools that can confer selective activation and/or suppression of distinct subsets of HDB neurons. The identification of the VGLUT3[+] subset of HDB neurons allows for the study of one specific projection to the MOB that regulates the GL-dSAC microcircuit. In future studies, VGLUT3[Cre] mice can be used to specifically address the impact of HDB cholinergic signaling and co-transmission in the IPL on circuit level processing and olfactory-guided behavior. Additionally, our observation that HDB projections release not only acetylcholine, but also GABA may help explain previous discrepancies in the study of HDB modulation of MOB function. For example, co-transmission such as we observed here would predict that cholinergic agonists and antagonists would mediate different effects on MOB activity and olfactory-guided behavior than activation or inactivation of HDB projections.

Our histology reveals the clear presence of GABAergic markers in cholinergic neurons in the HDB, consistent with recent reports of colocalization of GABAergic and cholinergic markers in the globus pallidus, medial septum, and other BF nuclei[31–33]. The lack of significant ionotropic glutamatergic currents in VGLUT3[Cre] mice may instead suggest a metabotropic role for VGLUT3, or reflect developmental down-regulation of the transporter, as has been observed in several neuronal populations elsewhere in the nervous system[34–38]. Lastly, the partial colocalization of VGAT with VAChT or ChR2-EYFP in the IPL may reflect a more restricted distribution of release sites for GABA compared to acetylcholine in these processes. Not all recorded MOB neurons showed evidence of GABAergic input together with nicotinic receptor activation. In addition, although there are clear reports of neurons in which multiple neurotransmitters are co-packaged into the same vesicles,

neurons showing segregation of neurotransmitters across release sites or even within the same neural process have also been reported[33, 39–42]. Factors affecting precisely how and where GABA and acetylcholine are synaptically released by HDB projections to the IPL, and whether VGLUT3 has a metabotropic or developmental role, will be important topics for future studies.

## Methods

**Animals**. Animals were treated in compliance with Institutional Animal Care and Use Committees for University of Pittsburgh, with the Declaration of Helsinki, and with the Institute for Laboratory Animal Research of the National Academy of Science's Guide for the Care and Use of Laboratory Animals. All efforts were made to minimize the number of animals used and to avoid pain or discomfort. Mice were housed in micro-isolator cages on a standard 12-h sleep/wake and were provided food and water ad libitum. Mice were typically injected with virus no earlier than 3 weeks of age, and were used by 7 weeks of age for electrophysiology experiments, and 10 weeks of age for anatomical experiments. The number of males and females used in each experiment were approximately equal. Mice were on a C57Bl/6 background except for VGLUT3[Cre], which was on a mixed C57Bl/6 and FVB/N background. ChAT[Cre] (stock #018957) and Ai14 tdTomato reporter mice (stock #007914) were obtained from Jackson Laboratories. VGLUT3[Cre] mice were described previously[23].

**Stereotaxic injections**. Mice at least 3 weeks of age (typically P21 for electrophysiological experiments; P21, P28, P35, or P42 for anatomical experiments) were anaesthetized by inhalation of 2.5% isoflurane, mounted in a stereotax, a small craniotomy was performed, and the mice were injected unilaterally or bilaterally with AAV9.EF1a.DIO.hChR2(H134R)-EYFP.WPRE.hGH (AAV9-DIO-ChR2-EYFP) (0.75–1 μL; Addgene 20298, supplied by Penn Vector Core) into the HDB at the following coordinates (in mm):+0.70 AP, ±0.85 ML, and −5.3 DV. The injection needle was lowered slowly to the target and was allowed to settle for at least 3 min. Virus was injected at a rate of 0.25 μL per minute, followed by 3-min incubation, and the injection needle was slowly removed. Following surgery, mice were given a mild painkiller (Ketofen, 5 mg/kg) and were returned to their home cage for recovery once they regained full activity. Mice recovered for a minimum of 2 weeks (maximum of 6 weeks) in order to allow sufficient viral expression. No differences in innervation pattern were observed following different lengths of expression within this time frame.

**Immunohistochemistry**. After a virus expression period of at least 2 weeks, mice were anesthetized with a lethal dose of a ketamine/xylazine mixture and transcardially perfused with ice-cold PBS (in mM: 137 NaCl, 2.7 KCl, 10.1 Na2HPO4, and 1.8 KH2PO4) followed by ice-cold PBS with paraformaldehyde (PFA; 4%). The brains were dissected from the animal and incubated overnight in 4% PFA at 4°C. The next morning brains were transferred to 30% sucrose solution at 4°C for cryoprotection. Following sufficient cryoprotection, 30 μm sagittal sections were cut on a cryostat (Microm HM550) and applied directly to glass slides or placed into wells. For fluorescent labeling, sections were incubated on slides or in wells with 5% normal donkey serum and 1% Triton X-100 in PBS for 1 h at room temperature (RT). Primary antibodies including goat anti-ChAT 1:1000 (Millipore AP144P); rabbit anti-VAChT 1:800 (Synaptic Systems #139 103); guinea pig anti-VAChT 1:500 (Synaptic Systems 139 105); rabbit anti-vesicular GABA transporter 1:1000 (Synaptic Systems 131 002), or rabbit anti-GFP 1:1000 (Invitrogen 11122) were diluted in the same solution and left on the sections overnight at 4°C. The next morning, sections were washed several times with PBS solution, followed by incubation in Alexa Fluor-conjugated secondary antibodies (Jackson ImmunoResearch) for 1 h at RT. Sections were washed several times and then underwent a final incubation in a dilute Sudan Black B solution to remove autofluorescence that can arise in the MOB. Slides were coverslipped with Fluoromount-G (Southern Biotech), and imaged by confocal microscopy.

**In situ hybridization**. The protocol as follows was described previously[37, 43]. Probe sequences were obtained from the Allen Brain Atlas and were generated by reverse transcription PCR from a brain cDNA library using the following primers: VGAT forward: 5′-gccattcagggcatgttc-3′, VGAT reverse: 5′-agcagcgtgaagaccacc-3′, VAChT forward: 5′-accccacagaaagtgaagatgt-3′, VAChT reverse: 5′-aagtgagtgaac-gatatggcct-3′, tdTomato forward: 5′-atcaaagagttcatgcgcttc-3′, tdTomato reverse: 5′-gttccacgatggtgtagtcctc-3′. Probes were generated using fluorescein-12-UTP and digoxigenin-UTP labeling mixes (Roche). Fresh frozen adult VGLUT3[Cre]-tdTomato brains were cut using a cryostat (Microm HM550) into 30 μm coronal sections, mounted on Superfrost slides (Fisher Scientific) and stored at −80°C. Sections were fixed in 4% PFA for 20 min, washed, subjected to proteinase K treatment, washed, fixed with 4% PFA, washed, deacetylated, subjected to hydrogen peroxide (0.3%), washed, and dehydrated through a series of ethanol concentrations. Probes were denatured at 85°C in hybridization buffer (50% formamide, 10% dextran sulphate, 200 μg/ml yeast tRNA, 1x Denhardts,

0.25% SDS, 600 mM NaCl, 10 mM Tris-HCl pH 8.0, and 1 mM EDTA pH 8.0) and then placed on the sections overnight at 65 °C. Sections were washed in decreasing concentrations of saline-sodium citrate buffer (5x to 0.1x) at 65 °C and then incubated with blocking buffer and then with a sheep anti-flu-horseradish peroxidase antibody (Roche) overnight at 4 °C. Next day, a sheep anti-dig-alkaline phosphatase antibody (Roche) was added for 3 hs at RT. Sections were washed and then subjected to tyramide amplification (NEN) and streptavidin-Alexa488 (Invitrogen) binding as well as the HNPP reaction kit (Roche) for fast red fluorescence. Slides were coverslipped in Vectashield (Vector) mounting media and imaged by epifluorescence or confocal microscopy.

**Confocal imaging**. Confocal images were taken on a Nikon Eclipse Ti-E inverted confocal with Nikon Elements software. For fluorescence-associated viral expression or fluorescently tagged antibodies, z-stack images were taken of the HDB and MOB using a 10x or 20x dry objective. For images in which fluorescence correlation was determined, single plane images of a sub-region of the IPL in the MOB were taken using a ×60 oil objective. Laser intensities were set for each image to minimize the number of both saturated and under-exposed pixels.

**Acute slice preparation**. Postnatal day 35–47 mice were anesthetized with isoflurane and decapitated into ice-cold oxygenated dissection solution containing (in mM): 125 NaCl, 25 glucose, 2.5 KCl, 25 NaHCO3, 1.25 NaH2PO4, 3 MgCl2, and 1 CaCl2. Brains were isolated and acute sagittal slices (310 μm thick) were prepared using a vibratome (VT1200S; Leica). Slices recovered for 30 min in ~ 37 °C oxygenated Ringer's solution that was identical to the dissection solution except for lower $Mg^{2+}$ concentrations (1 mM MgCl2) and higher $Ca^{2+}$ concentrations (2 mM CaCl2). Slices were then stored at RT until recording.

**Electrophysiology**. Slices were continuously superfused with 34 °C oxygenated Ringer's solution. Cells were visualized using infrared differential interference contrast video microscopy. Most recordings were made using electrodes filled with (in mM): 140 Cs-gluconate, 10 QX-314, 2 KCl, 10 HEPES, 10 Na-phosphocreatine, 4 Mg-ATP, 0.3 Na3GTP, 0.025 AF594, and 0.2% Neurobiotin (Vector Labs). A small subset of recordings were made using electrodes filled with (in mM): 120 K-gluconate, 2 KCl, 10 HEPES, 10 Na-phosphocreatine, 4 Mg-ATP, 0.3 Na3GTP, 0.2 EGTA, 0.025 AF594, and 0.2% Neurobiotin. Liquid junction potentials were 12–14 mV ($K^+$-based solution) and 11 mV ($Cs^+$-based solution) and were not corrected for. Data were low-pass filtered at 4 kHz and digitized at 10 kHz using a MultiClamp 700 A amplifier (Molecular Devices) and an ITC-18 acquisition board (Instrutech) controlled by custom software written in IGOR Pro (WaveMetrics). For optogenetic stimulation, slices were illuminated (10–20 ms light pulse) by a 75 W xenon arc lamp passed through an EYFP filter set and ×20–60 water-immersion objective centered on the recorded cell. NBQX (Tocris and Sigma), AP5 (Tocris), Gbz (Tocris), and Mec (Tocris and Alomone Labs) were added as described. Cell morphology was reconstructed under a ×100 oil-immersion objective and analyzed with Neurolucida (MBF Bioscience). Reconstructed dendrites and axons are drawn in black and red, respectively. Synaptic responses were classified as significant if the peak amplitude of the mean response across all trials exceeded 3 × SD of the baseline noise. Synaptic latencies were calculated as the interval from the onset of the light pulse onset to the time at which the synaptic response reached 5% of its peak amplitude. Normally and non-normally distributed data (Anderson-Darling test) were respectively analyzed using parametric and non-parametric tests, as described. All electrophysiological analyses were performed in MATLAB (MathWorks). Electrophysiological values are reported as mean ± SD; error bars and shading denote mean ± SEM. Traces have been down-sampled to 1 kHz for visualization.

**Data analysis and statistics**. Immunohistochemical experiments from un-injected mice were analyzed using >15 cells per slice and >4 slices from in most cases $n = 3$ animals. For injection experiments, we used at least 3 animals for each experimental condition. Due to the nature of the experiments, data collection and analysis did not require blinding or randomization. Nor did the experiments require special inclusion or exclusion criteria. To compare virus-associated fluorescence (AAV9-DIO-ChR2-EYFP) in the MOB between ChAT^Cre and VGLUT3^Cre mice injected in the HDB (Fig. 2), 10X z-stack images were taken in 3 sections of MOB from each animal (representing the lateral, middle, and medial thirds of the MOB). Maximum intensity projections were then thresholded to remove background fluorescence. Then, regions of interest were drawn and analyzed from each layer (GCL, IPL, EPL, and GL). The average fluorescence was calculated for each region and normalized to the fluorescence level of the IPL; the values from the 6 regions of interest in each layer (2 regions per layer in 3 sections) were averaged to determine a relative fluorescence value for each animal. No differences were observed in the relative fluorescence levels obtained when considering either the mediolateral or dorsoventral location. The average layer fluorescence values from a single animal were then averaged with the values from the other mice of the same genotype to create a population average.

Fluorescence correlation values for AAV9-DIO-ChR2-EYFP infected VGLUT3^+ projections (or ChAT^+ projections) and antibody labeled VAChT protein were calculated using the coloc2 function in ImageJ. Values reported are Pearson's R, which represent the level of overlap of the two fluorescent signals (proportion of EYFP signal overlapping the VAChT signal). Data shown represent the mean ± SEM and were analyzed by Prism version 5 software (GraphPad). Asterisks indicate statistical significance.

**Data availability**. The authors declare that data supporting the findings of this study are available within the paper and its supplementary information files and are also available from the corresponding author upon reasonable request.

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

## Acknowledgements

We thank Xinyi Zhao and Dhir Patwa for help with HDB injections, Charlene Yuan for help with immunostaining and confocal microscopy, Greg LaRocca for technical assistance, and Matt Wachowiak for providing the GAD2Cre-GCaMP6f tissue. Funding was provided by National Institute on Deafness and Other Communication Disorders Grants F31DC013490 (S.D.B.), R01DC005798 (N.N.U.), R01DC011184 (N.N.U.), the Pennsylvania Department of Health Commonwealth Universal Research Enhancement Program (N.N.U.), and National Institute of Neurological Disorders and Stroke R01NS082650 (R.P.S.).

## Author contributions

D.T.C. and R.P.S. conceived the study; all authors designed experiments and analyzed data; D.T.C. performed stereotaxic injections and immunostaining; S.D.B. performed electrophysiological recordings; J.G. performed HDB injections and immunostaining; S.P.G.W. performed immunostaining and in situ hybridization; D.T.C., S.D.B., N.N.U., and R.P.S. wrote the manuscript with contributions from J.G. and S.P.G.W.; D.T.C. and S.D.B. contributed equally to this work.

## Additional information

**Competing interests:** The authors declare no competing financial interests.

