## [Peer Review File · Nature Communications]

Reviewers' comments:

Reviewer #1 (Remarks to the Author):

This study examines the functional properties and olfactory bulb targeting of axons of a specific subset of cholinergic neurons in the basal forebrain that also express vesicular glutamate transporter (VGLUT) 3. The authors combined viral injection methods in different Cre mouse lines together with immunohistochemical and electrophysiological analyses. Evidence is provided, first, that the targeting of this subset of cholinergic fibers, which is specific to the internal plexiform layer (IPL), is much more selective than the general population of cholinergic fibers. The specificity also appears to extend to cell-type, since light activation of a population of VGLUT3-containing cholinergic fibers elicits currents in a subtype of deep short-axon cells (dSACs) that extend processes to the glomerular layer but not many other cell-types. Furthermore, based on pharmacological analyses, the currents appear to have cholinergic, glutamatergic, and GABAergic components. Based on these results, the authors conclude that neurotransmitter co-release from cholinergic fibers is important in the regulation of a specific inhibitory microcircuit in the olfactory bulb.

Overall, this is an interesting study that addresses important questions about the anatomy and function of cholinergic fibers that extend from the basal forebrain to the olfactory bulb. The data are of high quality and the results are generally well-presented. Much of the evidence supporting the target specificity of VGLUT3-containing cholinergic projections from basal forebrain is also generally convincing. My concerns are mainly with experiments suggesting co-release of neurotransmitters, which is the component that may be of broadest interest. The authors' evidence that VGLUT3-containing cholinergic fibers can release glutamate at all needs to be solidified. Also, the evidence supporting co-release of neurotransmitters (Ach, glutamate, and GABA) at the level of single cells and/or single synapses is now relatively modest.

Major Points

1. The evidence that VGLUT3-containing cholinergic fibers release glutamate is not convincing. Optogenetic activation of cholinergic/VGLUT3-containing fibers elicited an EPSC in dSACs that is reduced by only ~15% due to NBQX/APV and no reversal of the effect is shown. With such small decreases in current, it is important to demonstrate that the reduced current does not reflect simple run-down.

2. Better evidence for co-release of neurotransmitter needs to be provided. Figure 21 illustrates co-localization of VAcHT and VGLUT-associated td-Tomato, supporting that co-release of Ach and glutamate could happen at single synapses. However, the number of instances in which such co-localization was observed (3) is small, and the authors also report that they were unable to find co-localization between VAcHT and another marker for VGLUT3 (immuno-stained VGLUT3). The large number of instances in which neurotransmitter markers appear not to be co-localized would be consistent with most synapses not co-releasing multiple neurotransmitters.

The immunohistochemical data in Fig. 21 appears to show one instance in which markers for Ach, glutamate, and GABA co-localize, suggesting that co-release of Ach and GABA (along with glutamate) could occur at single synapses, but such evidence is quite preliminary. In addition, no other data supporting co-release of Ach and GABA at the level of single cells are provided.

The authors do provide evidence in Figure 1d,e that markers for VGLUT3 and Ach co-localize in single axons, supporting that a single cholinergic basal forebrain cell could release glutamate and Ach. However, the lack of convincing physiological data that the axons can release glutamate onto dSACs (see Major Point 1) reduces the impact of this result.

3. The evidence for cell type-specific targeting of the VGLUT3-containing cholinergic fibers should be strengthened. The evidence that these fibers target glomerular layer GL(glomerular layer)-dSACs at a high level is convincing and the authors do demonstrate lack of targeting onto a number of other cell-types in the bulb. However, more evidence that granule cell layer-projecting dSACs are not targeted ideally could be provided. The authors report that activation of VGLUT3-containing cholinergic fibers does not elicit currents in such dSACs with cell bodies in the IPL, but the number of such cells analyzed ($n = 3$) is relatively small, and no recordings were made from granule cell layer-projecting dSACs with cell bodies in the granule cell layer. Further recordings from granule cell layer-projecting dSACs will enable the authors to make stronger conclusions about the inhibitory microcircuit-specificity of the cholinergic axons.

Other Points

1. Some of the results now in Supplementary figures should be part of main figures. Particularly noteworthy in this regard are the morphological results showing that the dSACs that receive inputs from cholinergic/VGLUT3-containing fibers are a subtype that extends processes to the glomerular layer.

2. The authors suggest that the latencies to the nicotinic and glutamatergic currents (illustrated in Fig. 2j) are similar. It would help to show the traces on an expanded scale to make this point.

Reviewer #2 (Remarks to the Author):

The authors identify two basal forebrain cholinergic projections to the olfactory bulb (OB) and their putative neural targets. Approximately half of the diagonal band (DB) cholinergic neurons express (or more strictly at some point “expressed”) vglut3, consistent with potential glutamate/acetylcholine co-expression and/or co-transmission. The vglut3 expressing fibers heavily terminate in the internal plexiform layer where they selectively target a subpopulation of deep short axon cells (dSACs). The data presented detailing the two HDB populations and the highly selective targeting of the dSACs by the vglut3 is strong although several targeted neurons, designated “?-dSACs, did not have confirmed GL projections thus mitigating the author’s claim of selectivity.

The pharmacology indicates the Chr2 activation of BF vglut3 or ChAT-cre neurons triggers release of ACh, GABA, and glutamate upon dSACs. However, as the authors point out this could be due to separate BF cell populations that release one of the transmitters as well as BF neurons that express and release multiple transmitters. Indeed, it is also possible that different terminals of the same BF neurons could release either ACh or Glut, implying co-transmission but not necessarily co-release. The double label is suggestive of such multiple expression but the degree of overlap between the transporters/construct in the IPL is modest. Indeed, they state that when assessed by immunohistochemistry co-expression of vGlut and vAChT protein were not observed. Thus, the question of multiple expression vs. multiple populations is unresolved and reduces somewhat the potential impact of the report. As a result, the oft-stated conclusion that BF neurons co-release these transmitters is not warranted by conclusive evidence.

The abstract implies they show that the two BF cholinergic projections preferentially innervate GL or IPL. However, while they clearly show one population preferentially targets IPL, the conclusion that the other ACh population is preferential to the GL is an inference and not demonstrated by the authors’ findings. The second population could target IPL as well as GL, and the authors may prematurely dismiss the possibility that BF projections target neurons in other layers. The abstract should be reframed to indicate the demonstrated specificity for one BF population.

The electrophysiology is solid, as one would expect from this group of investigators. However,

with regard to ACh, the focus is entirely on nicotinic receptors. This may be somewhat short sighted as several studies have demonstrated a significant role for potent muscarinic receptor modulation of OB neurons and circuitry.

Minor points

The observation of light evoked EPSCs in the Chat-ChR2 BAC line is from one of two cells (lines 61-63). These data are insignificant to the paper and not of a sufficient sample set to warrant publication. The authors should either expand to a significant sample size or eliminate this observation from the report.

The Mander's coefficient is generally not considered an accurate co-localization metric. Considering that the rotated image in the vglut3 generates nearly half the Mander's value raises strong concern that the Mander's metric overestimates co-localization (one of the metrics weaknesses is that it over-emphasizes smaller numbers of bright points). A Pearson's coefficient would be a better choice for supplemental figure 2.

Reviewer #3 (Remarks to the Author):

In the manuscript by Case et al., the authors explored the possibility of mixed cholinergic-glutamatergic-GABAergic neurotransmitter co-release from HDB cells into the Olfactory Bulb (OB). By using a combination of molecular, pharmacological, optogenetic, and electrophysiological approaches, the authors convincingly demonstrate that both Chat and Vglut3 expressing neurons in the HDB are capable of multi-neurotransmitter release into the OB, specifically onto internal plexiform layer short axon cells. Although there are some concerns to be addressed, I believe this paper offers valuable insight to the field on how HDB projections may differentially regulate OB neurons through multi-neurotransmitter co-release, and should be further considered for publication.

Comments:

- The quantification of how many Vglut3 cells are ChAT positive is appreciated, but it would also be useful to know how many ChAT positive cells are Vglut3 positive. One realizes that viral infection rates may present a confound, but estimates here could be useful.
- In figure 2b, it's unclear why the authors utilized ChATCre/ChR2 mice to perform circuit mapping. It is not completely clear whether all (or some) of the optogenetic experiments are performed via injection of conditional ChR2 virus into the HDB, or if the ChAT-ChR2 line was

used exclusively for this. This is at least important to describe/differentiate, given the numerous cholinergic cell types throughout the brain. Also, to help the readers understand the area of infection, it would be useful to include a low magnification image of HDB injected region.

- In figure 2d, f, it's unclear why the authors used different pharmacological blockade experimental paradigms. For comparison purposes, it might benefit the readers to keep them the same, or describe why the different approaches were implemented.

- Supplementary Figure 1: The intensity of VACht staining makes co-localization difficult to see. A higher magnification image with cellular resolution may provide more convincing data.

- Smith et al., 2015 (J. Neuroscience) previously reported that optogenetic activation of HDB cholinergic projections stimulate Accessory Olfactory Bulb mitral cells, and inhibit Main Olfactory Bulb mitral cells. It would be useful to provide some commentary on similarities and discrepancies with the current work.

- While VACht stains synaptic terminals, a more definitive marker for cell body quantification is ChAT. Although not required for this reviewer, a possible experiment would be to inject AAV-DIO-ChR2-EYFP (or any AAV-DIO-FP as a cell fill) into the HDB of Vglut3 Cre animals and then stain for ChAT. This may help provide better quality images of colocalization.

- Although temporal dynamics provide strong evidence of direct connectivity, TTX addition alongside 4AP to evaluate postsynaptic currents (without the need for light activated APs) may provide further definitive data.

Response to Reviewers' comments:

We thank the reviewers for their thoughtful and insightful comments on our manuscript, which have improved it significantly. Taking these comments to heart, we have now made numerous revisions, including the introduction of new experiments, analyses, and clarifications. There are now 6 main figures and 2 supplementary figures (formerly 2 main figures and 5 supplementary figures).

In particular, we clarified and strengthened our data supporting:

- 1) Co-transmission of ACh and GABA by an HDB projection to the IPL
- 2) Selective innervation of GL-dSACs by this projection
- 3) A prominent role of nicotinic, but not muscarinic, cholinergic transmission onto GL-dSACs

In addition, we now clarify the issue of glutamate release as well as the issue of co-transmission versus co-release as specified in the responses below.

Reviewer #1 (Remarks to the Author):

We thank the strong support of reviewer 1.

Overall, this is an interesting study that addresses important questions about the anatomy and function of cholinergic fibers that extend from the basal forebrain to the olfactory bulb. The data are of high quality and the results are generally well-presented. Much of the evidence supporting the target specificity of VGLUT3-containing cholinergic projections from basal forebrain is also generally convincing.

Major Points

1. The evidence that VGLUT3-containing cholinergic fibers release glutamate is not convincing. Optogenetic activation of cholinergic/VGLUT3-containing fibers elicited an EPSC in dSACs that is reduced by only ~15% due to NBQX/APV and no reversal of the effect is shown. With such small decreases in current, it is important to demonstrate that the reduced current does not reflect simple run-down.

We agree with the reviewer that the manuscript left unclear whether the HDB-IPL projection elicits glutamate-mediated EPSCs. We did not find evidence for rundown during the recordings. Although the reduction in EPSC amplitude upon application of glutamate receptor blockers was consistent across cells, it did not reach significance. We also did not detect VGLUT3 protein in VAcHT⁺ fibers in the adult MOB by immunostaining. Though surprisingly, EPSCs have been detected at synapses that lack of immunoreactivity to VGLUTs (Weisz et al Nature, 2009, Seal et al Neuron 2008, Hnasko et al 2010) most likely because the levels are low. Nevertheless, based on the data we have obtained in this study, we conclude that glutamate release by HDB-IPL projections in the adult makes an

insignificant contribution to the EPSC and have now clarified this in our manuscript on lines 35, 119 and 181 and in Figure 4. Like many other VGLUT3⁺ populations, regulated glutamate release mediated by VGLUT3 may have a more prominent role during development (Boulland JL. et al 2004, Gras c et al 2005, Gillespie D. et al 2005; Peirs C et al 2015).

2. Better evidence for co-release of neurotransmitter needs to be provided. Figure 2l illustrates co-localization of VACHT and VGLUT-associated td-Tomato, supporting that co-release of Ach and glutamate could happen at single synapses. However, the number of instances in which such co-localization was observed (3) is small, and the authors also report that they were unable to find co-localization between VACHT and another marker for VGLUT3 (immuno-stained VGLUT3). The large number of instances in which neurotransmitter markers appear not to be co-localized would be consistent with most synapses not co-releasing multiple neurotransmitters.

In terms of VGLUT3, as mentioned above, we were unable to detect significant glutamate-mediated EPSCs nor were we able to detect VGLUT3 immunoreactivity in tdTomato⁺ axons in the IPL and we have now altered the text to better reflect our conclusion that we do not observe co-transmission of glutamate in the adult. We agree with the reviewer that our figure showing co-localization of VACHT and dtTomato could be improved and so we now provide clearer images of these two markers in the IPL in Figures 3 and 6.

Importantly, we have now expanded on our observation of GABA and ACh co-transmission by the HDB-IPL projection. New data presented in Figure 6 and the main text show significant colocalization of VGAT and VACHT (47%) and VGAT and tdTomato (43%) in HDB neurons of VGLUT3^{Cre}-tdTomato mice by in situ hybridization. We also report significant co-localization of ChAT and GCaMP6f in the HDB of GAD2^{Cre}-GCaMP6f mice (48%). Lastly, we have now included images in Figure 6 showing VACHT and VGAT colocalization in the MOB of Chr2-EYFP injected VGLUT3^{Cre} mice. (see also point 3)

3. The immunohistochemical data in Fig. 2l appears to show one instance in which markers for Ach, glutamate, and GABA co-localize, suggesting that co-release of Ach and GABA (along with glutamate) could occur at single synapses, but such evidence is quite preliminary. In addition, no other data supporting co-release of Ach and GABA at the level of single cells are provided.

We agree with the reviewer regarding ACh and GABA co-release vs. co-transmission and have revised the text to only describe the finding in terms of co-transmission. In addition, we now provide single cell level support for the co-expression of VGAT and VACHT in HDB neurons in Figure 6 and is consistent with a recent publication by Saunders et al who demonstrated a high degree of co-localization between ChAT and VGAT as well as ChAT and GAD2 in the MS/HDB (Saunders et al eLife, 2015).

4. The authors do provide evidence in Figure 1d,e that markers for VGLUT3 and Ach co-localize in single axons, supporting that a single cholinergic basal forebrain cell could release glutamate and Ach. However, the lack of convincing physiological data that the axons can release glutamate onto dSACs (see Major Point 1) reduces the impact of this result.

We agree with the reviewer as discussed in point 1 above.

The evidence for cell type-specific targeting of the VGLUT3-containing cholinergic fibers should be strengthened. The evidence that these fibers target glomerular layer GL(glomerular layer)-dSACs at a high level is convincing and the authors do demonstrate lack of targeting onto a number of other cell-types in the bulb. However, more evidence that granule cell layer-projecting dSACs are not targeted ideally could be provided. The authors report that activation of VGLUT3-containing cholinergic fibers does not elicit currents in such dSACs with cell bodies in the IPL, but the number of such cells analyzed (n = 3) is relatively small, and no recordings were made from granule cell layer-projecting dSACs with cell bodies in the granule cell layer. Further recordings from granule cell layer-projecting dSACs will enable the authors to make stronger conclusions about the inhibitory microcircuit-specificity of the cholinergic axons.

We focused most of our recordings on cells in or around the IPL and superficial GCL given the pronounced density of cholinergic and VGLUT3-positive projections we observed in this layer. Further recordings were not attempted in deeper layers given the sparser innervation patterns observed there. As noted by the reviewer, however, it is possible that dSACs located in deeper layers may receive cholinergic input from en passant synapses of basal forebrain axons as they project to the IPL. We have noted this possibility in our revised manuscript on line 96.

While our sample size is relatively small for GCL-dSACs, the lack of cholinergic input to these cells is quite striking, given that they were located in the IPL – a region of dense cholinergic innervation. Likewise, all of the GCs that we recorded were located in or near the IPL, and they also lacked significant cholinergic input.

Moreover, in our revised manuscript we have further analyzed the morphological properties of the 13 dSACs whose axonal projections could not be conclusively classified (?-dSACs). Of these 13, all exhibited nicotinic input, and of those, we were able to reconstruct the somatodendritic morphology of 5. These 5 ?-dSACs exhibited highly similar morphological properties as the confirmed GL-dSACs but tended to diverge from the morphological properties of the confirmed GCL-dSACs (Supplementary Table 1). In particular, GCL-dSACs tended to exhibit: 1) longer dendrites (reflected in greater total dendritic length and maximum Scholl radius), 2) deeper laminar distribution of dendrites, 3) substantially more dendritic spines, and 4) more complex dendritic arbors

(reflected in greater fractal indices and dendritic branching). Therefore, we suspect that the population of ?-dSACs was highly enriched for GL-dSACs, further suggesting that GL-dSACs – and not GCL-dSACs – may be selectively innervated by cholinergic basal forebrain projections that selectively innervate the IPL. This new morphological analysis and discussion has been added to our revised manuscript (Supplementary Table 1 and main text on lines 80-97).

Other Points

1. *Some of the results now in Supplementary figures should be part of main figures. Particularly noteworthy in this regard are the morphological results showing that the dSACs that receive inputs from cholinergic/VGLUT3-containing fibers are a subtype that extends processes to the glomerular layer.*

We agree with the reviewer and have moved the Supplementary Figure 1 into Figure 1 and supplementary Figure 2 into Figure 2 and Supplementary Figure 5 to Figure 5.

2. *The authors suggest that the latencies to the nicotinic and glutamatergic currents (illustrated in Fig. 2j) are similar. It would help to show the traces on an expanded scale to make this point.*

Given our new conclusions regarding glutamatergic currents (glutamate release makes an insignificant contribution to the EPSC in adult mice – see above discussion), we have removed discussion and waveforms of glutamatergic current latencies from our text and figure, respectively. However, in agreement with the reviewer’s general suggestion, we have now displayed average nicotinic EPSC and GABAergic IPSC waveforms (Fig. 4i), which clearly shows equal latencies between the two signals.

Reviewer #2 (Remarks to the Author):

We appreciate the strong support of this reviewer.

1. The authors identify two basal forebrain cholinergic projections to the olfactory bulb (OB) and their putative neural targets. Approximately half of the diagonal band (DB) cholinergic neurons express (or more strictly at some point “expressed”) vglut3, consistent with potential glutamate/acetylcholine co-expression and/or co-transmission. The vglut3 expressing fibers heavily terminate in the internal plexiform layer where they selectively target a subpopulation of deep short axon cells (dSACs). The data presented detailing the two HDB populations and the highly selective targeting of the dSACs by the vglut3 is strong although several targeted neurons, designated “?-dSACs, did not have confirmed GL projections thus mitigating the author’s claim of selectivity.

In our revised manuscript, we have further analyzed the morphological properties of the 13 dSACs whose axonal projections could not be conclusively classified (?-dSACs). Of these 13, all exhibited nicotinic input, and of those, we were able to reconstruct the somatodendritic morphology of 5. These 5 ?-dSACs exhibited highly similar morphological properties as the confirmed GL-dSACs but tended to diverge from the morphological properties of the confirmed GCL-dSACs (Supplementary Table 1). In particular, GCL-dSACs tended to exhibit: 1) longer dendrites (reflected in greater total dendritic length and maximum Scholl radius), 2) deeper laminar distribution of dendrites, 3) substantially more dendritic spines, and 4) more complex dendritic arbors (reflected in greater fractal indices and dendritic branching). Therefore, we suspect that the population of ?-dSACs was highly enriched for GL-dSACs, further suggesting that GL-dSACs – and not GCL-dSACs – may be selectively innervated by cholinergic basal forebrain projections that selectively innervate the IPL. This new morphological analysis and discussion has been added to our revised manuscript (Supplementary Table 1 and main text on lines 80-97).

2. The pharmacology indicates the ChR2 activation of BF vglut3 or ChAT-cre neurons triggers release of ACh, GABA, and glutamate upon dSACs. However, as the authors point out this could be due to separate BF cell populations that release one of the transmitters as well as BF neurons that express and release multiple transmitters. Indeed, it is also possible that different terminals of the same BF neurons could release either ACh or Glut, implying co-transmission but not necessarily co-release. The double label is suggestive of such multiple expression but the degree of overlap between the transporters/construct in the IPL is modest. Indeed, they state that when assessed by immunohistochemistry co-expression of vGlut and vAChT protein were not observed. Thus, the question of multiple expression vs. multiple populations is unresolved and reduces somewhat the potential impact of the report. As a result, the oft-stated conclusion that BF neurons co-release these transmitters is not warranted by conclusive evidence.

We agree with the reviewer about the nature of the release and have modified the manuscript to address this by providing additional data in Figures 3 and 6 and by altering the language in the text from co-release to co-transmission. In Figure 6, we now show colocalization of ACh and GABA markers in the HDB by in situ hybridization and immunostaining. Additionally in Figure 3, we have added more data on colocalization of cholinergic and GABAergic markers in the IPL. We have also included a discussion on the nature of the release in lines 184-188. The modest overlap of both VACHT and VGAT in the ChR2-EYFP fibers could be due to many factors including issues with detection and segregation of the transmitters within the axons. Saunders et al (2015) recently reported a GP-to-cortical circuit showing ACh and GABA co-transmission. In this case individual axons contained mixed synapses: some with both transmitters, some with only one or the other. Segregation of transmitters within processes of the same neurons or the same fibers has been reported for other neurons that show co-transmission (Zhang S et al, Nat Neuro, 2015, Sethuramanujam S et al., Neuron, 2016, Lee, S et al Neuron, 2010, Saunders et al, Nature 2015).

Despite the present but limited overlap of GABA and ACh markers in ChR2-EYFP-expressing axons, light-evoked postsynaptic currents mediated by GABA and ACh show equivalent latencies consistent with monosynaptic events elicited from these ChR2-EYFP fibers (Figure 4 and lines 124-125). Moreover, the GABAergic currents were not blocked by the cholinergic antagonist mec, further supporting co-transmission versus a feed-forward mechanism. Lastly, the fact that both GABAergic and nicotinic postsynaptic currents occur in both ChAT^{Cre} and VGLUT3^{Cre} mice makes it highly likely that the cholinergic projection in the VGLUT3^{Cre} mice releases GABA. This is further supported by the observation that 6 out of 6 neurons from VGLUT3^{Cre} mice and 4 out of 8 neurons from ChAT^{Cre} showed a GABAergic postsynaptic current (Figure 4 and lines 129-130).

3. The abstract implies they show that the two BF cholinergic projections preferentially innervate GL or IPL. However, while they clearly show one population preferentially targets IPL, the conclusion that the other ACh population is preferential to the GL is an inference and not demonstrated by the authors' findings. The second population could target IPL as well as GL, and the authors may prematurely dismiss the possibility that BF projections target neurons in other layers. The abstract should be reframed to indicate the demonstrated specificity for one BF population.

We agree with the reviewer that our data does not rule out that cholinergic projections from neurons not expressing VGLUT3^{Cre} also innervate the IPL. We have therefore revised the title and text, including the abstract.

3. The electrophysiology is solid, as one would expect from this group of investigators. However, with regard to ACh, the focus is entirely on nicotinic

receptors. This may be somewhat short sighted as several studies have demonstrated a significant role for potent muscarinic receptor modulation of OB neurons and circuitry.

As stated by the reviewer, several studies across multiple laboratories have utilized muscarinic agonists to demonstrate potent modulation of many MOB neuron types, such as MCs and GCs, through either direct or indirect actions. Of note, however, two recent studies from the Shipley and Araneda laboratories failed to observe strong muscarinic modulation of MCs when using channelrhodopsin to evoke release of endogenous ACh. Liu et al. (2015) observed no significant effect in MCs following a brief light pulse (similar to our own approach and results), while Smith et al. (2015) used a 15 s-long train of light pulses (50 ms pulses at 10 Hz) and observed only a 0.5 mV hyperpolarization in MCs on average. Thus, while muscarinic receptors are clearly present throughout the MOB, they may only be activated under specific circumstances involving prolonged ACh release.

In our recordings, we did not observe any evidence of long-lasting Mec-insensitive currents mediated by muscarinic receptors following brief light pulses (10-20 ms) in either dSACs or other MOB cell types (Fig. 4d,f; Fig. 5n), including a subset of cells recorded with a K⁺-based intracellular solution (Fig. 5b,e) that should not attenuate any muscarinic modulation of K⁺ channels. This lack of light-evoked muscarinic currents is consistent with the above recent reports of no muscarinic modulation of MCs following brief optogenetic stimulation of BF cholinergic projections¹¹, or very weak hyperpolarization (~0.5 mV) following prolonged (15-s 10 Hz train) optogenetic stimulation⁴.

Our revised manuscript text and figures reflect this discussion of muscarinic currents (lines 107 to 115).

Minor points

The observation of light evoked EPSCs in the Chat-ChR2 BAC line is from one of two cells (lines 61-63). These data are insignificant to the paper and not of a sufficient sample set to warrant publication. The authors should either expand to a significant sample size or eliminate this observation from the report.

We agree with the reviewer and have eliminated the data from the ChAT-ChR2 transgenic mouse line from the manuscript.

The Mander's coefficient is generally not considered an accurate co-localization metric. Considering that the rotated image in the vglut3 generates nearly half the Mander's value raises strong concern that the Mander's metric overestimates co-localization (one of the metrics weaknesses is that it over-emphasizes smaller numbers of bright points). A Pearson's coefficient would be a better choice for

supplemental figure 2.

We thank the reviewer for pointing this out and have now re-analyzed the data in Supplementary Figure 1 (formerly Supplementary Figure 2) using calculations of Pearson's coefficient.

Reviewer #3 (Remarks to the Author):

We appreciate the strong support of Reviewer 3.

In the manuscript by Case et al., the authors explored the possibility of mixed cholinergic-glutamatergic-GABAergic neurotransmitter co-release from HDB cells into the Olfactory Bulb (OB). By using a combination of molecular, pharmacological, optogenetic, and electrophysiological approaches, the authors convincingly demonstrate that both Chat and Vglut3 expressing neurons in the HDB are capable of multi-neurotransmitter release into the OB, specifically onto internal plexiform layer short axon cells. Although there are some concerns to be addressed, I believe this paper offers valuable insight to the field on how HDB projections may differentially regulate OB neurons through multi-neurotransmitter co-release, and should be further considered for publication.

Comments:

1. The quantification of how many Vglut3 cells are ChAT positive is appreciated, but it would also be useful to know how many ChAT positive cells are Vglut3 positive. One realizes that viral infection rates may present a confound, but estimates here could be useful.

We have analyzed the number of ChAT⁺ cells that are VGLUT3⁺ (~46%) and now report it at the beginning of the results section (line 47). We also included a discussion of the numbers of ChAT⁺ neurons that were infected with the virus in each mouse line in the legend for Figure 2.

2. In figure 2b, it's unclear why the authors utilized ChATCre/ChR2 mice to perform circuit mapping. It is not completely clear whether all (or some) of the optogenetic experiments are performed via injection of conditional ChR2 virus into the HDB, or if the ChAT-ChR2 line was used exclusively for this. This is at least important to describe/differentiate, given the numerous cholinergic cell types throughout the brain. Also, to help the readers understand the area of infection, It would be useful to include a low magnification image of HDB injected region.

We agree with the reviewer (and reviewer 2) that data from the ChAT-ChR2 mouse line would require more detailed characterization, but is too sparse to warrant inclusion. We have therefore removed the data from this mouse line from the manuscript.

3. In figure 2d, it's unclear why the authors used different pharmacological blockade experimental paradigms. For comparison purposes, it might benefit the readers to keep them the same, or describe why the different approaches were implemented.

We have clarified our motivation for the different pharmacological paradigms in

the revised text (pgs 4 and 5). In short, NBQX/AP5/Gbz was combined in recordings from VGLUT3^{Cre} mice to investigate beyond any doubt whether these projections release acetylcholine. Therefore, we blocked all other major modes of possible ionotropic synaptic transmission. The finding that optogenetic stimulation of VGLUT3⁺ projections prior to the addition of NBQX/AP5/Gbz also evoked an IPSC was unexpected, and thus not systematically examined using the prolonged sequential pharmacology experiment used in Fig. 4f and Fig. 4g right.

4. *Supplementary Figure 1: The intensity of VAcHT staining makes co-localization difficult to see. A higher magnification image with cellular resolution may provide more convincing data.*

We thank the reviewer for pointing this out and have moved the data from Supplementary Figure 1 into Figure 1 and have uniformly increased the intensity for easier visualization.

5. *Smith et al., 2015 (J. Neuroscience) previously reported that optogenetic activation of HDB cholinergic projections stimulate Accessory Olfactory Bulb mitral cells, and inhibit Main Olfactory Bulb mitral cells. It would be useful to provide some commentary on similarities and discrepancies with the current work.*

As noted by the reviewer, the Araneda laboratory (as well as the Shipley laboratory) recently examined optogenetic activation of BF cholinergic projections to the MOB. The Shipley laboratory observed no significant effect in MCs following a brief light pulse (similar to our own approach and results) (Liu et al., 2015), while the Araneda laboratory used a 15 s-long train of light pulses (50 ms pulses at 10 Hz) and observed only a 0.5 mV hyperpolarization in MCs on average (Smith et al., 2015). Thus, though muscarinic receptors are clearly present throughout the MOB, they may only be activated under specific circumstances involving prolonged ACh release. These results are directly consistent with our own findings. Specifically, we did not observe any evidence of long-lasting Mec-insensitive currents mediated by muscarinic receptors following brief light pulses (10-20 ms) in either dSACs or other MOB cell types, including MCs (Fig. 4d,f; Fig. 5n).

This comparison to previous results from the Araneda and Shipley laboratories, as well as evaluation of muscarinic currents, is now included in our revised manuscript text and figures.

6. *While VAcHT stains synaptic terminals, a more definitive marker for cell body quantification is ChAT. Although not required for this reviewer, a possible experiment would be to inject AAV-DIO-ChR2-EYFP (or any AAV-DIO-FP as a cell fill) into the HDB of Vglut3 Cre animals and then stain for ChAT. This may help provide better quality images of colocalization.*

We thank the reviewer for bringing this to our attention. As we did for images in Supplementary Figure 1 (now Figure 1), we improved the clarity of images that were in Supplementary Figure 3 and have moved them into Figure 2.

7. Although temporal dynamics provide strong evidence of direct connectivity, TTX addition alongside 4AP to evaluate postsynaptic currents (without the need for light activated APs) may provide further definitive data.

As stated by the reviewer, the short latencies of the light-evoked responses strongly suggest a monosynaptic connection from basal forebrain to GL-dSACs in the MOB. In addition, nicotinic responses persisted after pharmacological blockade of ionotropic glutamatergic and GABAergic transmission (Fig. 4d,f,g). This finding, together with the lack of intrinsic cholinergic neurons within the MOB, argues against the possibility that basal forebrain activation is driving nicotinic input to GL-dSACs via disynaptic feedforward input. Likewise, GABAergic responses persisted after pharmacological blockade of ionotropic glutamatergic and nicotinic responses (Fig. 4f,g), confirming monosynaptic GABAergic input. Therefore, while the suggested TTX/4AP experiment would provide a complementary test of our conclusion of monosynaptic connectivity between basal forebrain and GL-dSACs, we feel that this additional experiment is not currently necessary.

Reviewers' Comments:

Reviewer #1 (Remarks to the Author):

In the revised manuscript, the authors have some nice additions, including more evidence for co-transmission of GABA and Ach from VGLUT+ cholinergic fibers. They also are no longer making a significant claim about co-transmission of glutamate and ACh, which addresses many of the prior concerns that I raised. The point about the target selectivity of the VGLUT+ cholinergic fibers however remains ambiguous.

Remaining Concerns

1. The evidence that VGLUT+ cholinergic fibers selectively target GL-dSACs over GCL-dSACs is still modest. I raised the concern previously that the number of sampled GCL-dSACs ($n = 3$) was small. In response, the authors have added an analysis of 13 “?-dSACs” in which axonal projections of the neurons could not be determined. They argue based on the dendritic morphology that 5 of the cells were likely to be GL-dSACs rather than GCL-dSACs. The data (shown in Supp Table 1) are however not convincing. The authors cite a number of dendritic morphological parameters for which the ?-dSACs had values different from those of the confirmed GCL-dSACs. This would suggest that the ?-dSACs were not GCL-dSACs. However, the parameter values for “?-dSACs” were generally no more similar to those for GL-dSACs. In addition, there was one parameter (total dendritic volume) in which the values for “?-dSACs” were similar to those for GCL-dSACs but not for GL-dSACs. In my opinion, the analysis of the ?-dSACs around this point is not helpful and should be removed. Ideally, the authors would perform more recordings in cells confirmed to be GCL-dSACs or they should at least tone down their claim that the VGLUT+ cholinergic fibers selectively target GL-dSACs over GCL-dSACs.

2. The evidence that VGLUT+ cholinergic fibers are distinct from VGLUT- fibers in that they specifically target the IPL needs to be strengthened. The summary of the fluorescence data in Figure 2e makes a strong point, but the images shown in Figure 2c do not. The fluorescence in the IPL is not robust, and there appears to be fluorescence in the GL as well as part of the GCL. The authors should show the ROIs used in the analysis of these images and also a plot detailing the fluorescence estimates arrived at from these particular images. The authors should also be clear how they accounted for the fact that within the GL, fluorescence appears to be restricted to regions around glomeruli rather than within glomeruli. This would naturally lead to lower overall fluorescence estimates for the GL versus IPL, even while the fluorescence in the regions around glomeruli could be reasonably high. It is not enough that the same issues should have impacted the fluorescence estimates in the ChAT-Cre mice.

The authors could also strengthen the electrophysiological evidence that the VGLUT+ cholinergic fibers do not extend to the GL. The summary plot in Figure 5 indicates that the authors did not observe currents in PG cells and eTCs (both in the GL), but currents were only measured in single examples of each cell-type.

Other Points

1. In Fig. 6d and e, it appears that VGAT and VAcHT only infrequently co-localize. Does this mean that most axons that release both VGAT and VAcHT do so at different synapses? This needs to be discussed. GABA and ACh would likely have opposing effects (inhibition versus excitation) on their post-synaptic partners, and so a situation in which GABA and ACh are generally not co-released at the same synapse could be important.
2. For the inset of Fig. 4i, no scale bar is shown.
3. The histogram bars on the right-hand side of Fig. 4g indicate that Mec reduced the light-evoked GABAergic current in ChAT-Cre mice by 60%. This would argue that a major component of the inhibition is feedforward (and not just direct). This needs to be pointed out.

Reviewer #2 (Remarks to the Author):

The authors have thoughtfully responded to the previous concerns and revised the manuscript accordingly. As a result the paper is much improved and, IMO, suitable for acceptance.

Reviewer #3 (Remarks to the Author):

The authors have thoughtfully addressed reviewer suggestions/concerns. I endorse publication of the revised manuscript in Nature Communications.

Response to the Reviewers

We thank all three reviewers for their helpful comments, which have significantly improved our manuscript.

Reviewer #1 (Remarks to the Author):

In the revised manuscript, the authors have some nice additions, including more evidence for co-transmission of GABA and Ach from VGLUT+ cholinergic fibers. They also are no longer making a significant claim about co-transmission of glutamate and ACh, which addresses many of the prior concerns that I raised. The point about the target selectivity of the VGLUT+ cholinergic fibers however remains ambiguous.

We appreciate the strong support of this reviewer and have addressed remaining concerns below.

Remaining Concerns

1. The evidence that VGLUT+ cholinergic fibers selectively target GL-dSACs over GCL-dSACs is still modest. I raised the concern previously that the number of sampled GCL-dSACs (n = 3) was small. In response, the authors have added an analysis of 13 “?-dSACs” in which axonal projections of the neurons could not be determined. They argue based on the dendritic morphology that 5 of the cells were likely to be GL-dSACs rather than GCL-dSACs. The data (shown in Supp Table 1) are however not convincing. The authors cite a number of dendritic morphological parameters for which the ?-dSACs had values different from those of the confirmed GCL-dSACs. This would suggest that the ?-dSACs were not GCL-dSACs. However, the parameter values for “?-dSACs” were generally no more similar to those for GL-dSACs. In addition, there was one parameter (total dendritic volume) in which the values for “?-dSACs” were similar to those for GCL-dSACs but not for GL-dSACs. In my opinion, the analysis of the ?-dSACs around this point is not helpful and should be removed. Ideally, the authors would perform more recordings in cells confirmed to be GCL-dSACs or they should at least tone down their claim that the VGLUT+ cholinergic fibers selectively target GL-dSACs over GCL-dSACs.

The reviewer raises excellent points, and upon further review of our data, we agree with the reviewer that we cannot conclude with certainty that EPL-dSACs and/or at least a subset of GCL-dSACs are not likewise innervated by cholinergic HDB projections. We have integrated this amended conclusion throughout our revised manuscript. In addition, we have removed Supp. Table 1 from our revised manuscript.

2. The evidence that VGLUT+ cholinergic fibers are distinct from VGLUT- fibers in that they specifically target the IPL needs to be strengthened. The summary of the fluorescence data in Figure 2e makes a strong point, but the images shown in Figure 2c do not. The fluorescence in the IPL is not robust, and there appears to be fluorescence in the GL as well as part of the GCL. The authors should show the ROIs used in the analysis

of these images and also a plot detailing the fluorescence estimates arrived at from these particular images. The authors should also be clear how they accounted for the fact that within the GL, fluorescence appears to be restricted to regions around glomeruli rather than within glomeruli. This would naturally lead to lower overall fluorescence estimates for the GL versus IPL, even while the fluorescence in the regions around glomeruli could be reasonably high. It is not enough that the same issues should have impacted the fluorescence estimates in the ChAT-Cre mice.

We appreciate the reviewer's comment that we needed to better clarify our point about the targeting of VGLUT3^{Cre} HDB cholinergic fibers. As the reviewer points out and as illustrated by the graph in Fig 2f, these fibers are clearly densest in the IPL, but are also present in GCL (and EPL). This differs significantly from the innervation pattern observed for the ChAT^{Cre} cholinergic fibers to the MOB, which are also very dense in IPL, but also in GL and are present to a more significant degree in the EPL. Hence, we have now altered the manuscript to better highlight this point and we also now show clearer representative images (two for VGLUT3^{Cre} fibers) in Fig 2c-e and we also include additional representative images in a new Supplementary Figure 1.

The authors could also strengthen the electrophysiological evidence that the VGLUT+ cholinergic fibers do not extend to the GL. The summary plot in Figure 5 indicates that the authors did not observe currents in PG cells and eTCs (both in the GL), but currents were only measured in single examples of each cell-type.

Other Points

- 1. In Fig. 6d and e, it appears that VGAT and VACHT only infrequently co-localize. Does this mean that most axons that release both VGAT and VACHT do so at different synapses? This needs to be discussed. GABA and ACh would likely have opposing effects (inhibition versus excitation) on their post-synaptic partners, and so a situation in which GABA and ACh are generally not co-released at the same synapse could be important.*

We agree that the co-localization of VACHT with VGAT is relatively low and we have discussed this on page 8 (copied below). There are many potential explanations for this observation based on what has been reported for other examples of co-release (see in particular Saunders A et al *Nature* 2015). First, not all HDB to IPL cholinergic fibers release GABA (page 5). Second, it may be the case that VACHT forms en passant synapses whereas VGAT and VACHT are colocalized only at the nerve terminal (hence segregation of the transporters within the neurons). Lastly, it is also possible that VGAT expression in cholinergic fibers is present at low levels and hence underrepresented by antibody staining as has been reported for VGLUT2 in both DA neurons and in primary sensory neurons. Most of the VGAT in the MOB is derived from the large numbers of local neurons, which may express the transporter at higher

levels. Further studies will be aimed at more fully understanding the location, consequence and regulation of co-transmitter release by these neurons.

On page 8: “Lastly, the partial colocalization of VGAT with VAcHt or ChR2-EYFP in the IPL may reflect a more restricted distribution of release sites for GABA compared to acetylcholine in these processes. In addition to examples of neurons in which multiple neurotransmitters are co-packaged into the same vesicles, neurons showing segregation of neurotransmitters across release sites or even within the same neural process have also been reported^{28,33-35}. Factors affecting precisely how and where GABA and acetylcholine are synaptically released by HDB projections to the IPL, and whether VGLUT3 has a metabotropic or developmental role, will be important topics for future study.”

2. *For the inset of Fig. 4i, no scale bar is shown.*

We have now added a scale bar to the inset of Fig. 4i.

3. *The histogram bars on the right-hand side of Fig. 4g indicate that Mec reduced the light-evoked GABAergic current in ChAT-Cre mice by 60%. This would argue that a major component of the inhibition is feedforward (and not just direct). This needs to be pointed out.*

Fig. 4g summarizes our pharmacological results for EPSCs ($V_{\text{hold}} = -60$ mV) only. We have clarified this by better labeling the y-axis of Fig. 4g in our revised manuscript. The ~60% reduction in EPSC peak following mecamylamine application reflects the strong contribution of nicotinic input to the recorded cells. That mecamylamine does not yield a more complete reduction of the EPSC peak reflects the continued presence of an incompletely voltage-clamped GABAergic current (e.g., see Fig. 4f, blue trace). That this GABAergic component persists following NBQX/AP5 and mecamylamine application, but is abolished by subsequent gabazine application, provides strong evidence for direct release of GABA, rather than acetylcholine- or glutamate-driven feedforward inhibition. This explanation is included on page 5 of our revised manuscript.

Reviewer #1 (Remarks to the Author):

The authors have adequately addressed my prior concerns, and the study is now suitable for publication. The new images in Fig. 2c-e in particular are considerably more convincing.